# Automatic Masseter Muscle Accurate Segmentation from CBCT Using Deep Learning-Based Model

**DOI:** 10.3390/jcm12010055

**Published:** 2022-12-21

**Authors:** Yiran Jiang, Fangxin Shang, Jiale Peng, Jie Liang, Yi Fan, Zhongpeng Yang, Yuhan Qi, Yehui Yang, Tianmin Xu, Ruoping Jiang

**Affiliations:** 1Department of Orthodontics, Peking University School and Hospital of Stomatology, Beijing 100081, China; 2National Clinical Research Center for Oral Diseases, Beijing 100081, China; 3National Engineering Laboratory for Digital and Material Technology of Stomatology, Beijing Key Laboratory of Digital Stomatology, Peking University School and Hospital of Stomatology, Beijing 100081, China; 4NHC Research Center of Engineering and Technology for Computerized Dentistry, Beijing 100081, China; 5Intelligent Healthcare Unit, Baidu, Beijing 100081, China; 6Department of Oral and Maxillofacial Surgery, Peking University School and Hospital of Stomatology, Beijing 100081, China

**Keywords:** deep learning, machine learning, CBCT, masseter muscle, orthodontic(s), canio-maxillofacial surgery

## Abstract

Segmentation of the masseter muscle (MM) on cone-beam computed tomography (CBCT) is challenging due to the lack of sufficient soft-tissue contrast. Moreover, manual segmentation is laborious and time-consuming. The purpose of this study was to propose a deep learning-based automatic approach to accurately segment the MM from CBCT under the refinement of high-quality paired computed tomography (CT). Fifty independent CBCT and 42 clinically hard-to-obtain paired CBCT and CT were manually annotated by two observers. A 3D U-shape network was carefully designed to segment the MM effectively. Manual annotations on CT were set as the ground truth. Additionally, an extra five CT and five CBCT auto-segmentation results were revised by one oral and maxillofacial anatomy expert to evaluate their clinical suitability. CBCT auto-segmentation results were comparable to the CT counterparts and significantly improved the similarity with the ground truth compared with manual annotations on CBCT. The automatic approach was more than 332 times shorter than that of a human operation. Only 0.52% of the manual revision fraction was required. This automatic model could simultaneously and accurately segment the MM structures on CBCT and CT, which can improve clinical efficiency and efficacy, and provide critical information for personalized treatment and long-term follow-up.

## 1. Introduction

As one of the strongest pairs of masticatory muscles, the masseter muscle (MM), plays an irreplaceable role in mastication and occlusal function [1]. Meanwhile, the MM supports the mandibular angle area, which has a noticeable impact on craniofacial morphology and external facial appearance [2,3,4]. 

Previous studies have revealed that the volume of the MM changes markedly within the entire course of orthodontic-orthognathic treatment, which affects the postsurgical stability, masticatory function, and patient’s external appearance [2,4,5,6,7,8,9]. Moreover, investigators have also observed the atrophy of MM and surrounding soft tissue during orthodontic treatment in female adults, which may lead to an unaesthetic appearance as a sign of aging [3,10]. Furthermore, the evaluation of MM function and morphology is also useful in understanding the mechanism of facial asymmetry [11], as well as in improving orthodontic treatment and in determining a correct retention period after treatment [12,13]. Therefore, the availability of a patient-specific three-dimensional (3D) MM model from imaging data that provides information about the orientation, size, and shape would be extremely useful for the planning of oral and maxillofacial surgery and orthodontic treatment as well as for deriving biomarkers for treatment monitoring and treatment response. 

The MM could be accurately segmented from computed tomography (CT) [2,4,5,7,8,9], and magnetic resonance imaging (MRI) [14]. However, the hard tissue is barely visualized in the MRI, and metal implants and restorations as well as orthodontic brackets produce severe artifacts. Moreover, CT has the disadvantage of a high radiation dose, which is prohibitive in the diagnosis and treatment of non-surgical patients. In contrast, cone-beam computed tomography (CBCT) has the advantages of a low radiation dose, fast imaging reconstruction speed, high spatial precision, and low cost [15]. Currently, CBCT has replaced CT and MRI in most stomatological practices (such as orthodontics, periodontics, alveolar surgery, and oral implantation), and its resolution of dentoskeletal structures can be even higher than traditional CT. 

However, CBCT has been only reported to analyze the 2D cross-sectional area of the MM [16], as well as the orientation and length measured by landmarks [13], but few 3D studies have been reported [3] because robust and accurate segmentation of the MM from CBCT is difficult due to the lack of a sufficient soft-tissue contrast to differentiate the MM structures from their background structures. Moreover, the voxel range of CBCT that includes the complete MM region and can be used for 3D reconstruction is usually between 0.125 mm and 0.3 mm. Manual segmentation needs to handle hundreds of cross-sectional images, which is very time-consuming and laborious [17]. Therefore, if there is a method to accurately segment the MM on CBCT, this can provide critical information for clinical stomatological practice as well as reduce the use of high radiation doses of CT. As depicted in Figure 1, although the variance in the Hounsfield unit (HU) values of the MM region in CBCT was greater than that in CT, there is a similar distribution of the HU values in MM regions and non-MM regions between CBCT and CT. This suggests the feasibility of applying the semantic segmentation model to segment the MM from CBCT.

In recent years, deep learning-based models, such as convolutional neural networks (CNNs), have shown promising results in various medical imaging segmentation tasks, which can reduce the workload as well as limit the variation in manual segmentation. In a review of the literature, Iyer et al. introduced a 2.5D DeepLabV3+ structure for masticatory muscle segmentation [18]. Chen et al. applied a 3D U-Net structure [19]. Qin et al. developed a feature-enhanced 3D nested U-Net [20]. All the above reports are based on CT data. As for the segmentation study in CBCT, previous studies have reported the achievement of various deep learning-based methods on accurate segmentation of maxillofacial hard tissues (such as maxilla and mandible, alveolar bone, teeth and mandibular canal, etc.) [21,22,23], but there are relatively few research results for soft tissue segmentation, meaning that the rich information on soft tissues has not been fully excavated. To date, researchers have used adversarial approaches to generate pseudo-MRI or pseudo-CT from CBCT to produce potentially more accurate soft tissue segmentations than CBCT alone [17,24,25]. However, the basis of these models is matching the output to the distribution of the target domain rather than the anatomy structures, which can easily produce randomized outputs or hallucinate anatomies, especially using unpaired imaging data [26], and can cause accumulated error and increase segmentation error. Moreover, the computational power needed to train this architecture increases dramatically, because four subnetworks need to be trained (a feature extractor, generator, discriminator, and segmentation model) [17].

To the best of our knowledge, there are no reliable and accurate methods to segment the MM on CBCT. Therefore, in this study, for the sake of exploring 3D information of the MM on CBCT for clinical study and clinical practice, we proposed and validated a deep-learning-based automatic segmentation model for CBCT under the refinement of paired CT.

## 2. Materials and Methods

### 2.1. Ethical Considerations

The study was conducted in accordance with the Declaration of Helsinki (as revised in 2013). The study was approved by the institutional review board of Peking University School and Hospital of Stomatology (PKUSSIRB-201944062) and individual consent for this retrospective analysis was waived.

### 2.2. Image Acquisition

Because patient collection was retrospective, two CBCT units were used: (1) NewTom VG (Quantitative Radiology), with exposure parameters of 110 kVp and 2–3 mA and an FOV of 15 cm × 15 cm; and (2) i-CAT FLX (Imaging Sciences International, Inc.), with exposure parameters of 120 kVp and 5 mA and an FOV of 16 cm × 13 cm. The voxel size was 0.3 × 0.3 × 0.3 mm^3^. The CT images were acquired by a 64-row spiral CT scanner (Philips Inc., Andover, MA, USA) at 120 kVp, 230 mA, and 1.0 mm layer thickness. The CT resolution was set to 512 × 512 × 236, and the voxel size was 0.4 × 0.4 × 1 mm^3^.

In total, 50 independent CBCT and 42 paired CT and CBCT (268 MM in total) were collected in this retrospective study for model training, validating and blind testing. The CT was considered as the reference.

### 2.3. CBCT to CT Superimposition

Paired CBCT and CT acquired from the same patients were collected. CBCT was performed for the orthodontic 3D cephalometric analysis and for assessment of alveolar bone thickness and the tooth movement boundary. Due to the incomplete scanning of the cranial base in the CBCT, which is the main reference to determine the position of the maxilla and mandible in orthognathic surgery, spiral CT images were performed presurgically for 3D digital orthognathic surgical planning. The average interval was 2.53 ± 2.23 months. CBCT and CT images were superimposed in Dolphin3D imaging software (version 11.8; Dolphin Imaging and Management Solutions, Chatsworth, CA, USA) (Figure 2) and were exported with orientation.

### 2.4. Manual Annotation of the MM

Manual annotation and adjustment were performed in ITK-SNAP 3.8.0 (http://www.itksnap.org (accessed on 19 December 2022)). 

Two observers (YR. J., observer 1; JL. P., observer 2) with 10 and seven years of clinical experience conducted the manual operation under the guidance of an expert in oral and maxillofacial anatomy (L. J) with more 15 years of surgical experience. The 50 independent CBCT were first automatically segmented [17] and underwent layer-by-layer manual adjustment. Fifteen pairs of CT and CBCT were randomly selected to form the blind test set and were annotated by both observers (Figure 3).

The manual annotation of the MM on the reference CT was set as the ground truth.

### 2.5. MM Auto-Segmentation

The flowchart of our automatic segmentation method is shown in Figure 4.

Because the soft tissues in CBCT are hard to observe, we designed a progressive 3D semantic segmentation network from coarse-grained to fine-grained segmentation from low resolution to high resolution. To adequately capture features at every scale, the recently proposed ReSidual U-block (RSU) [27] module, which is a tiny U-shaped network in a nutshell, is applied to construct the proposed MM segmentation network.

As illustrated in Figure 5, the encoders and decoders were constructed by the RSU module. The vision features resolution was down-sampled by 2× with every encoder and up-sampled by 2× with every decoder. The encoders transformed the images into vision features, which was convenient for the decoders to extract the segmentation results, and the different scale features were decoded progressively to output the full-resolution prediction. To embed the coarse-grained features (global features) into the low-resolution stages and fine-grained features (local features) into the high-resolution, the training paradigm named *deep supervision* was applied, which collected decoder outputs at every scale, down-sampled the ground truth to fit the resolution of the output, and generated the supervised signal (gradient) to adjust the network parameters.

#### Implement Details

We found that the HU values in the MM region were similar between CBCT and CT images, which allowed us to develop an automatic model that could be used for the MM auto-segmentation on both images. In detail, the HU values in the CBCT and CT images were clipped ranges from −325 to 400. The lower bound and upper bound were acquired by the 0.5 and 99.5 quantile of the foreground pixels, which has been a widely used strategy proposed by nnU-Net [28].

The proposed network consisted of several RSU modules with four stages to balance the cost of the computational resources and segmentation accuracy. The RSU modules were constructed by 3D operations (convolution, normalization, upsampling, etc.). The activation functions in the network were LeakyReLU to extend the response range of the feature maps [29].

The training procedure consisted of 20,000 iterations with two samples in every batch, which aimed to minimize the dice loss [30] and cross-entropy loss between the ground truth and model prediction. The weights of the loss functions were equivalent. During training, the SGD optimizer with momentum was applied. The learning rate increased from 0 to 0.001 in the initial 300 iterations (linear warm-up strategy), and decayed to 0 in the remaining iterations following the cosine curve. Only the standard data augmentation strategies were introduced including random 3D rotation, random resizing and cropping. It should be noted that the random flip strategy was not used to avoid the obfuscation of the left side and right side of the MM.

### 2.6. Evaluation of Geometric Accuracy of the Segmentations

We applied the Dice similarity coefficient (DSC) and average Hausdorff distance (aHD) to detect morphological and positional deviations. The DSC was defined as follows:(1)$${\mathrm{DSC}}_{\mathrm{Seg}}=\frac{2\left|A_{\mathrm{Seg}}\cap M_{\mathrm{Seg}}\right|}{\left|A_{\mathrm{Seg}}\right|+\left|M_{\mathrm{Seg}}\right|}$$
(2)$${\mathrm{DSC}}_{\mathrm{Back}}=\frac{2\left|A_{\mathrm{Back}}\cap M_{\mathrm{Back}}\right|}{\left|A_{\mathrm{Back}}\right|+\left|M_{\mathrm{Back}}\right|}$$
(3)$${\mathrm{DSC}}_{\mathrm{Mean}}=\frac{1}{2}\left({\mathrm{DSC}}_{\mathrm{Seg}}+{\mathrm{DSC}}_{\mathrm{Back}}\right)$$
where A_Seg_ and M_Seg_ are the auto-segmentation and manual annotation, respectively, and A_Back_ and M_Back_ are the backgrounds outside each segmentation. |A∩M| is the overlap area and |A|+|M| is the total area.

The aHD calculates the average distances of a point in A ($a\in A_{\mathrm{Seg}}$) to its closest point in M ($m\in M_{\mathrm{Seg}}$), and $\left|A_{\mathrm{Seg}}\right|$ is the total number of A, given as:(4)$$\mathrm{aHD}\left(A_{\mathrm{Seg}},M_{\mathrm{Seg}}\right)=\frac{1}{\left|A_{\mathrm{Seg}}\right|}\underset{a\in A_{\mathrm{Seg}}}{\sum }\underset{m\in M_{\mathrm{Seg}}}{\min}\left(\left\|a-m\right\|\right)\;$$

### 2.7. Clinical Suitability

We further collected 10 additional CBCT and CT (CBCT, n = 5, CT, n = 5, 10 × 2 MMs) out of the above datasets to investigate the clinical suitability of the model by computing the DSCs, aHDs and revised fractions between the original auto-segmentations and the oral and maxillofacial anatomy expert-revised auto-segmentations. 

### 2.8. Statistical Analysis

A descriptive analysis of the study variables was done, calculating the mean, standard deviation (SD) and 95% confidence intervals (95%CI) for all the continuous variables by SPSS software (version 25.0; IBM, Armonk, NY, USA). The Bland–Altman method was performed for the auto-segmentation performance between CBCT and CT. A paired *t*-test was used to compare the difference between manual annotations and auto-segmentations on CBCT. Statistical significance was defined as *p* < 0.05.

## 3. Results

### 3.1. Interobserver Variations

The interobserver variability between two observers is shown in Table 1. For CBCT images, the DSC_Mean_ was 96.05 ± 2.46% (95%CI 94.69, 97.42) and the mean aHD was 4.31 ± 1.31 mm (95%CI 3.52, 5.09). For CT images, the DSC_Mean_ and the mean aHD were 95.82 ± 1.52% (95%CI 94.98, 96.66) and 3.22 ± 1.14 mm (95%CI 2.59, 3.86), respectively. This indicates that the high agreement of manual annotations can be reached in both CBCT and CT images.

### 3.2. Comparison of Manual Annotation Difference between CBCT and CT

The DSC between CBCT manual annotations and its CT manual counterparts were 84.75 ± 3.76% and 85.82 ± 3.25% for the left and right MM, respectively, and the mean DSC considering background similarity (DCS_Mean_) was 90.16 ± 2.23%. The mean aHD was 5.41 ± 1.63 mm (Table 1). The results demonstrated that the human eye recognition of the MM anatomic structure in CBCT was limited, resulting in systematic differences between CBCT and CT manual annotations.

### 3.3. Model Performance

The model performance is summarized in Table 2. The DSC between CBCT auto-segmentations and its CT manual counterparts were 91.56 ± 0.97% (95%CI 91.20, 91.92) and 90.94 ± 1.33% (95%CI 90.44, 91.44) for the left and right MM, respectively, and DCS_Mean_ was 94.15 ± 0.68% (95%IC 93.90, 94.40). The mean aHDs were 3.68±1.01 mm (95%IC 3.30, 4.06). The backgrounds of the automatic results completely overlapped with the ground truth (mean ${\mathrm{DSC}}_{\mathrm{Back}}$ = 99.96%). 

In addition, the DSC_Mean_ between CT auto-segmentations and manual annotations was 94.45 ± 0.80% (95%IC 94.15, 94.75), and the mean aHDs was 3.67 ± 1.25 mm (95%IC 2.59, 3.86). 

A good agreement between CBCT and CT auto-segmentation was demonstrated in Figure 6. The Bland–Altman plots showed the mean ± SD (95%CI) of difference between CBCT and CT auto-segmentation DSC_Mean_, DCS_Left MM_, DSC_Right MM_, aHD_Mean_, aHD_Left MM_, aHD_Right MM_ against the ground truth as −0.299 ±0.472% (95%CI −1.224, 0.626), −0.284 ± 0.689% (95%CI −1.633, 1.066), −0.610 ± 1.064% (95%CI −2.696, 1.476), 0.009 ± 0.641 mm (95%CI 1.248, 1.266), −0.126 ± 0.883 mm (95%CI −1.605, 1.353) and 0.143 ± 0.992 (95%CI −1.801, 2.088). All differences in the DCS_Mean_ were within the 95%CI. Only 3.33% (1/30) difference of DCS_Left MM_ and DCS_Right MM_ were out of the 95%CI, and 6.67% (2/30) difference of all aHD-related measurements were out of the 95%CI. Moreover, the DSC_Mean_ and the mean aHD between CBCT and CT auto-segmentation results were 94.48 ± 0.74% and 2.42 ± 0.33 mm, respectively (Table 2). 

Table 3, paired t-test results have demonstrated a statistically significant improvement in auto-segmentation results compared with manual annotations (*p* < 0.05).

### 3.4. Time Cost

The average time cost of the manual annotations on one case were measured to be 1879.80 ± 338.80 s for CBCT and 2245.80 ± 531.60 s for CT, respectively. As for the automatic model, the average time cost was 5.64 ± 0.63 s for CBCT and 6.76 ± 0.76 s for CT, which is 332.22 to 333.30 times shorter than the human operation (Table 4). The acceleration in the segmentation speed greatly improves clinical efficiency and frees doctors’ hands.

### 3.5. Clinical Suitability

The clinical suitability results are summarized in Table 5. The manual revision was mainly focused on the superior extents (examples depicted in Figure 7a,b). A mean revision fraction of 0.52 ± 0.44% was needed for both CBCT and CT auto-segmentations, and the revision fraction ranged from 0.25% to 0.72% for each side of the MM. The mean DSC_Mean_ was 99.84 ± 0.14%, and the mean aHD was 0.92 ± 0.88 mm. SD, standard deviation.

## 4. Discussion

Masticatory muscles, especially the MM, play an irreplaceable role in the function and morphology of the human stomatognathic system. Numerous studies have shown that the MM is closely correlated with stability after orthognathic surgery, esthetics of frontal facial profile, the development of malocclusion and facial deviation, and occlusal stability after orthodontic treatment [2,3,4,5,6,7,8,9,10,11,12,13]. Therefore, to assist the surgical planning, as well as orthodontic diagnosis, treatment, and retention, it is important to obtain patient-specific 3D MM models by segmenting them from imaging data that provide comprehensive information about the orientation, size, and shape of the MM. 

CBCT is a widely available diagnostic method in stomatological practice with substantial advantages over CT and MRI examinations in terms of low radiation, fast imaging and cost efficiency. At present, the application of CBCT is mainly focused on the analysis of dentoskeletal hard structures. However, the abundant 3D information on soft tissues, especially masticatory muscles, has not been fully explored because the blurry resolution and fuzzy boundaries of soft tissue images in CBCT make the manual segmentation of masticatory muscles time-consuming and error-prone [17,23,24]. In this study, we demonstrated that high interobserver consistency can be reached in both the CBCT and CT manual annotation task. This may be attributed to the high proficiency of the MM anatomy of two observers as stomatologists under the guidance of an expert in oral and maxillofacial anatomy, as well as the selection of high-quality imaging data [19]. 

However, our study has shown an obvious discrepancy between manual annotations on pairwise CBCT and CT (the mean DSC = 90.16 ± 2.23%, and DCS of segmentation less than 90%, Table 1). In other words, the visual recognition of the MM structure on CBCT by the human naked eye was limited. Furthermore, since the operator could only segment recognizable portions of the masseter muscle, the low quality of the CBCT soft tissue imaging brings mainly systematic errors to manual annotations, while the random errors are relatively fewer. 

There have been numerous efforts to improve the soft tissue imaging quality and segmentation accuracy of CBCT, ranging from hardware improvement to software improvement. However, they are limited by the high computational complexity to perform the correction [31]. Recently, deep learning-based models have achieved unprecedented advances in various biomedical image segmentation tasks [17,18,19,20,21,22]. Among these, the U-Net architecture has demonstrated a dominant performance by combining both low-resolution information (for objective recognition) and high-resolution information (for accurate segmentation positioning) [32]. Extending 2D to 3D structures is preferable because it can effectively exploit the 3D spatial and structural information directly from the volumetric images [33]. On the other hand, due to the fuzzy boundary and complex gradient of biomedical images, deep architecture and high-resolution information are essential for more accurate segmentation [27]. The recent state-of-the-art U^2^-Net is proposed as a two-level U-shaped structure with a deeper architecture while maintaining high resolution feature maps at affordable memory and computational costs [27]. In this study, we validated the accuracy of a 3D version of the U^2^-Net. With high-quality CT images as a reference, the segmentation performance on CBCT by this deep-learning algorithm is not only comparable to CT auto-segmentations (Figure 7), but also is significantly better than the manual annotations on CBCT (the DSC_Mean_ exceeded 94%, and the DSC_seg_ exceeded 90%, *p* < 0.05, Table 2 and Table 3). Therefore, by learning the ground truth of paired CT manual segmentations, deep learning methods can not only expand the boundaries of human eye recognition, but also build a more accurate MM model.

With regard to the model performance on CT, although limited cases of CT data were trained, the mean DSC_Mean_ exceeded 94% in this study. Since the HU values in the MM region were similar between CBCT and CT images (Figure 1), we can use the two kinds of training data to construct an automatic model capable of performing the segmentation task on both images. Furthermore, our results have surpassed those of previous reports [18,19], which may be attributed to the self-developed progressive 3D semantic segmentation network that could capture adequate features at every scale. On the other hand, a higher level of interobserver agreement was achieved in this study compared with the results by Chen et al. [19], which could ensure the quality of data for training the model. As noted by Feng et al., a model does not learn from its mistakes in the way that humans do, and providing poor training data will be disadvantageous for the training results [34]. However, our results were slightly inferior to those of Qin et al. [20], which may be due to the overfitting of the model caused by data augmentation and the cross-validation method in the latter study. 

Moreover, in order to evaluate whether the auto-segmentation results misidentified other structures, we also calculated the similarity of the background. The results showed that the background of the auto-segmentations was highly consistent with its manual counterparts. In other words, our model somewhat under-segmented the MM, which may be caused by the fuzzy boundary of the MM and the statistical clipping of the MM HU range based on the nnU-Net strategy (the foreground pixels out of the 0.5–99.5 quantile range would be clipped) [27].

With regard to the clinical suitability of the model, minor manual modifications for both the CBCT and CT auto-segmentation results were needed in this study (the mean manual revision fraction was 0.52 ± 0.44%, Table 2). A previous study reported a mean 22.86% of case revisions [18], suggesting an obvious improvement in terms of accuracy of our proposed method. Refinement was focused on the superior extents (Figure 7a,b). The explanation was that visualization of the MM superior ligament is poor [1], and the model threshold settings affect the identification results.

Prior to clinical implementation, it is also important to determine whether these auto-segmentation results fall within the interobserver variability. In this study, statistical analyses showed that the auto-segmentations were less variable than the manual annotations (the standard deviations for both DSC and aHD of the auto-segmentation results were smaller than the corresponding interobserver variations), indicating that this automatic method can feasibly reduce interobserver variability. A consistent result was also reported in previous studies [18,19].

Although high MM segmentation accuracy from both CT and CBCT was achieved, some limitations of our study still exist. Firstly, a larger dataset and more diverse CT and CBCT with variable exposure conditions will be expected to improve the generality of our model. Therefore, we will continue to collect the pairwise dataset in the future. Secondly, the performance of the proposed algorithm has not been compared with the other publications due to the lack of widely accepted benchmarks. Our next work will propose such benchmarks.

## 5. Conclusions

Our proposed automatic model achieved a breakthrough in the segmentation of the MM on CBCT. Its performance was comparable to that of clinical experts segmenting CT images and was significantly superior to that of clinical experts segmenting CBCT images. The annotation labor cost was reduced by more than 332 times, and only minor manual modifications were required. With the help of proper software, this method would extend the CBCT application and improve clinical efficacy and efficiency as well as reduce the radiation exposure of personalized treatment and long-term follow-up.

## Figures and Tables

**Figure 1 jcm-12-00055-f001:**
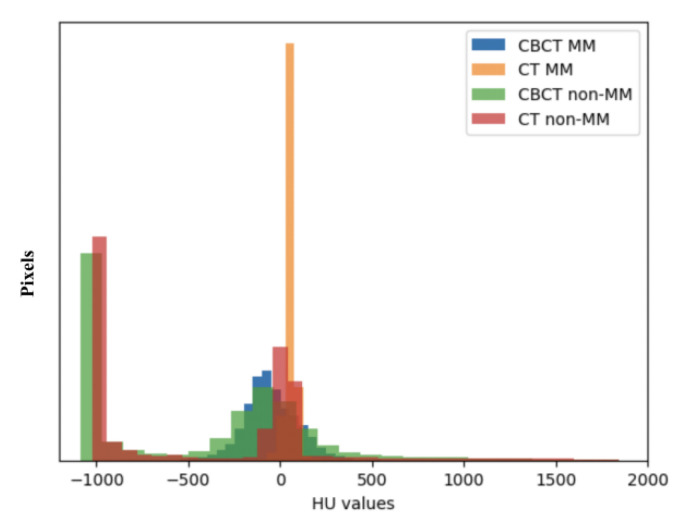
Hounsfield unit (HU) values distribution of the masseter muscle (MM) and non-masseter muscle (non-MM) regions in cone-beam computed tomography (CBCT) and computed tomography (CT).

**Figure 2 jcm-12-00055-f002:**
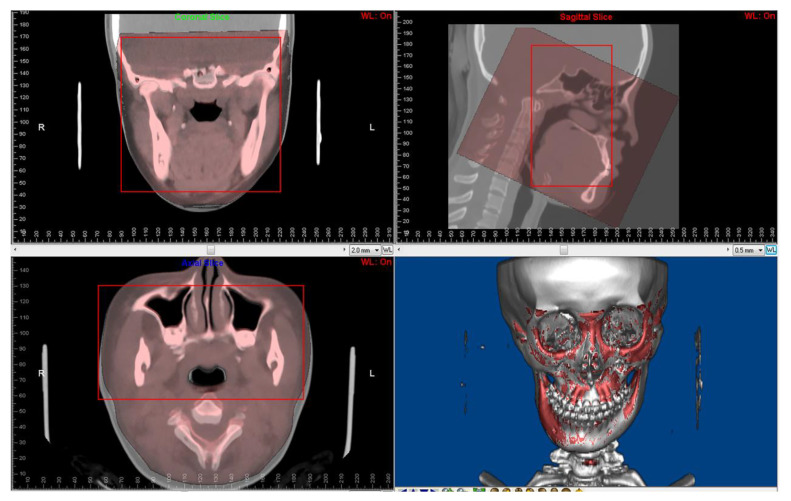
Superimposition of paired CT and CBCT was performed using Dolphin3D imaging software. The red squares are regions for superimposition.

**Figure 3 jcm-12-00055-f003:**
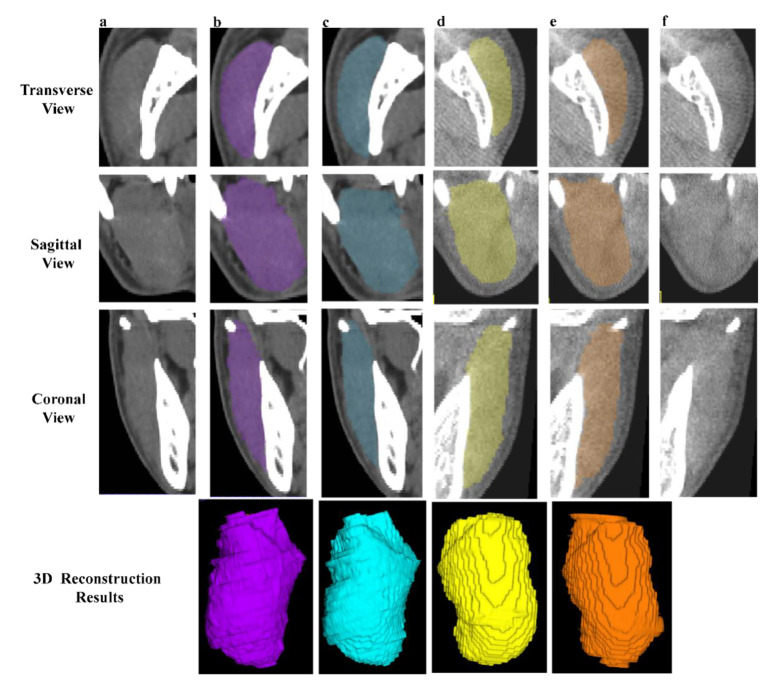
Examples of manual segmentation on CT and CBCT. (**a**,**f**) The original CT and CBCT images of right and left MM. (**b**,**e**) The manual annotations and 3D results of right (purple) and left (orange) MM by observer 1. (**c**,**d**) The manual annotations and 3D reconstruction results of right (blue) and left (yellow) MM by observer 2.

**Figure 4 jcm-12-00055-f004:**
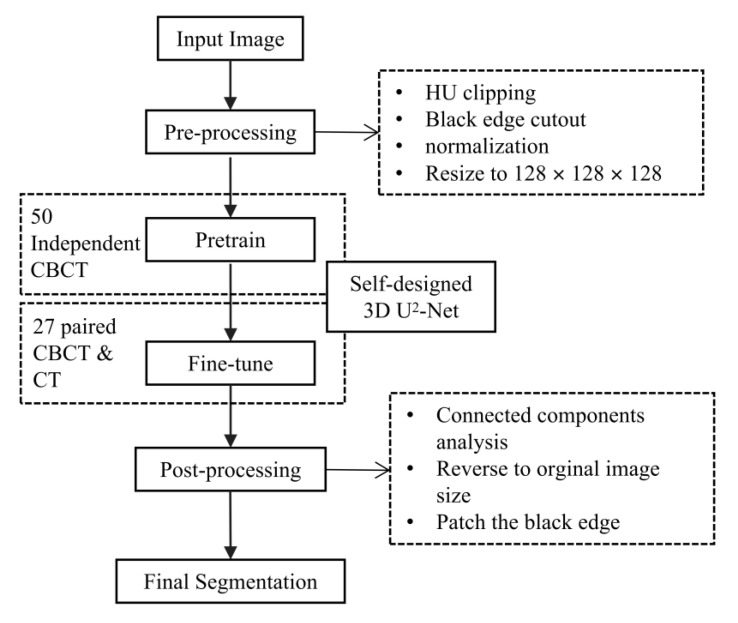
The flowchart of the proposed automatic segmentation method. CBCT, cone-beam computer tomography. CT, computer tomography. 3D, three-dimensional.

**Figure 5 jcm-12-00055-f005:**
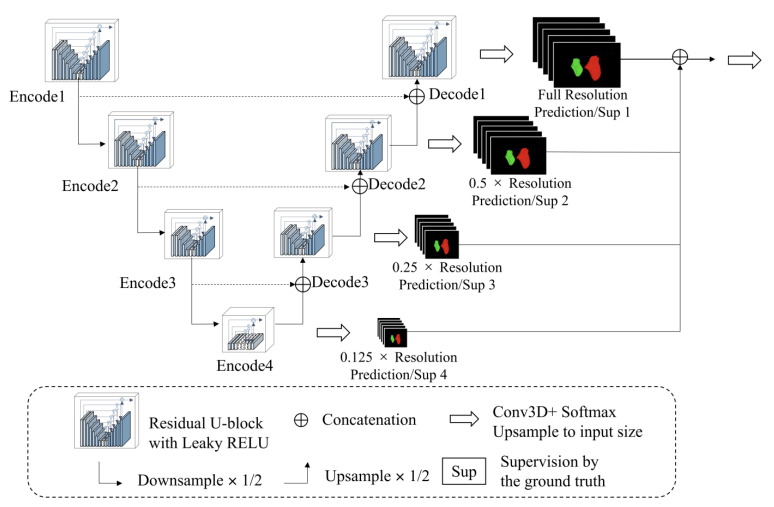
The architecture of the proposed model.

**Figure 6 jcm-12-00055-f006:**
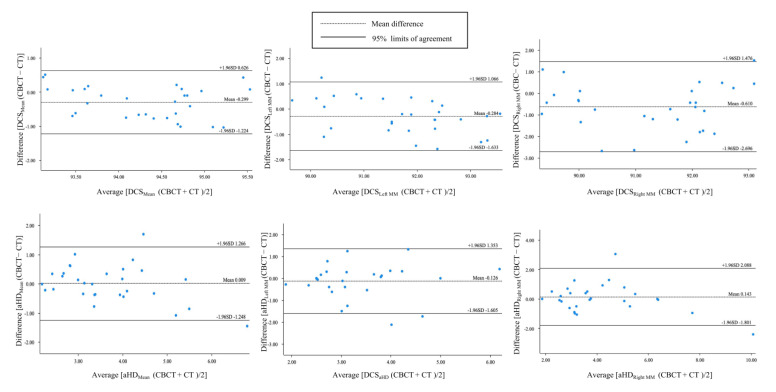
The Bland–Altman plots for agreement analysis between auto-segmentation results on CBCT and CT by the Dice similarity coefficient (DSC) and average Hausdorff distance (aHD).

**Figure 7 jcm-12-00055-f007:**
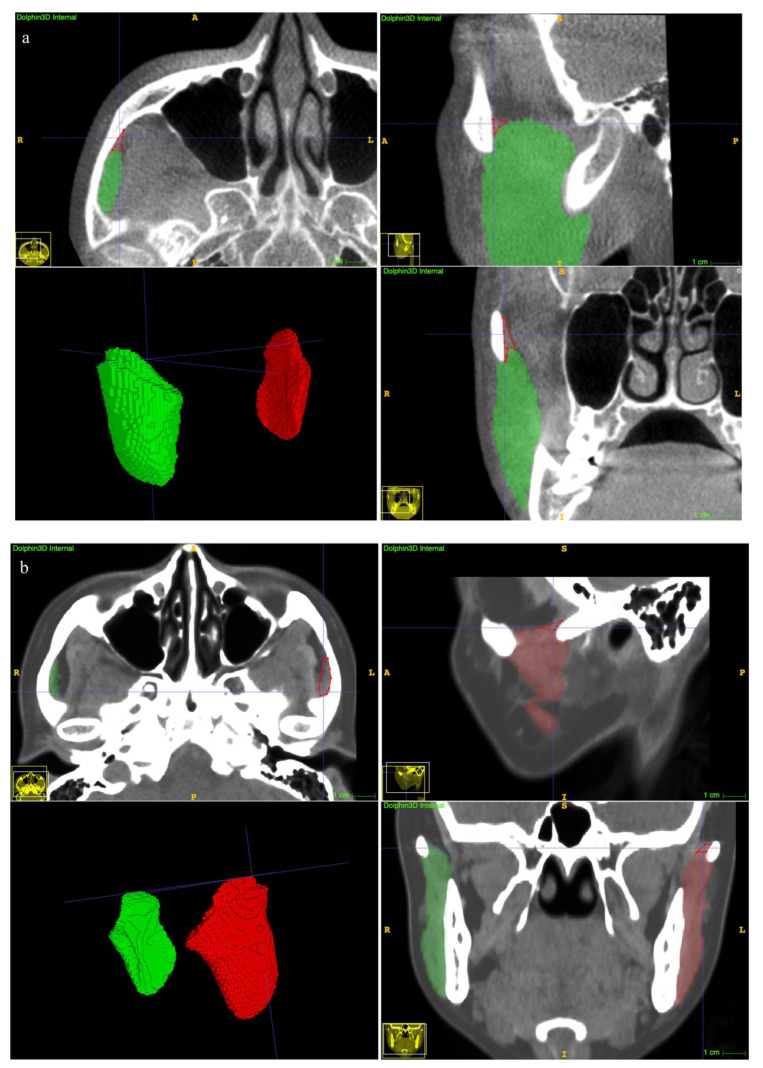
Examples of undersegmentation of the automatic model on the superior extent of the MM. The red polygon shows the undersegmented area. (**a**) CBCT segmentation results. (**b**) CT segmentation results.

**Table 1 jcm-12-00055-t001:** The interobserver variations and manual annotation difference between CBCT and CT by Dice similarity coefficient (DSC, %) and average Hausdorff distance (aHD, mm) among 15 cases.

	Mean ± SD (95%CI)						
Measurements	Dice Similarity Coefficient (DSC, %)				Average Hausdorff Distance (aHD, mm)		
	Mean	Background	Left MM	Right MM	Mean	Left MM	Right MM
Interobserver variations (CBCT)	96.05 ± 2.46 (94.69, 97.42)	99.97 ± 0.02 (99.96, 99.98)	94.39 ± 4.20 (92.07, 96.72)	93.78 ± 3.97 (91.59, 95.98)	4.31 ± 1.31 (3.52, 5.09)	4.36 ± 2.92 (2.74, 5.98)	4.26 ± 1.31 (3.53, 4.98)
Interobserver variations (CT)	95.82 ± 1.52 (94.98, 96.66)	99.97 ± 0.02 (99.96, 99.98)	93.95 ± 2.48 (92.58, 95.32)	93.57 ± 2.22 (92.31, 95.76)	3.22 ± 1.14 (2.59, 3.86)	2.99 ± 1.03 (2.33, 3.47)	3.55 ± 1.61 (2.66, 4.44)
CBCT manual annotations vs. CT manual annotations	90.16 ± 2.23 (89.33, 91.00)	99.93 ± 0.02 (99.92, 99,94)	84.75 ± 3.76 (83.22, 84.80)	85.82 ± 3.25 (84.61, 87.03)	5.41 ± 1.63 (4.80, 6.02)	5.28 ± 2.34 (4.41, 6.16)	5.54 ± 1.56 (4.95, 6.12)

SD, standard deviation; MM, masseter muscle; vs., versus.

**Table 2 jcm-12-00055-t002:** The evaluation of model performance by Dice similarity coefficient (DSC, %) and average Hausdorff distance (aHD, mm) among 15 cases.

	Mean ± SD (95%CI)						
Measurements	Dice Similarity Coefficient (DSC, %)				Average Hausdorff Distance (aHD, mm)		
	Mean	Background	Left MM	Right MM	Mean	Left MM	Right MM
CBCT auto-segmentations vs. CT manual annotations	94.15 ± 0.68 (93.90, 94.40)	99.96 ± 0.01 (99.95, 99.96)	91.56 ± 0.97 (91.20, 91.92)	90.94 ± 1.33 (90.44, 91.44)	3.68 ± 1.01 (3.30, 4.06)	3.22 ± 1.01 (2.84, 3.59)	4.14 ± 1.68 (3.52, 4.77)
CT auto-segmentations vs. CT manual annotations	94.45 ± 0.80 (94.15, 94.75)	99.96 ± 0.01 (99.96, 99.96)	91.84 ± 1.26 (91.37, 91.86)	91.55 ± 1.31 (91.06, 92.04)	3.67 ± 1.25 (3.21, 4.14)	3.35 ± 1.00 (2.97, 3.72)	4.00 ± 2.00 (3.25, 4.74)
CBCT auto-segmentations vs. CT auto-segmentations	94.48 ± 0.74 (94.07, 94.90)	99.96 ± 0.01 (99.95, 99.97)	91.89 ± 1.23 (91.21, 92.57)	91.60 ± 1.22 (90.93, 92.27)	2.42 ± 0.33 (2.24, 2.60)	2.27 ± 0.38 (2.06, 2.49)	2.57 ± 0.55 (2.26, 2.87)

SD, standard deviation; MM, masseter muscle; vs., versus.

**Table 3 jcm-12-00055-t003:** Paired *t*-test for evaluation difference between auto-segmentations and manual annotations on CBCT.

Measurements		Mean ± SD		Mean Difference (1–2)	*t*	*p* Value
		1:CBCT Auto-Segmentation	2:CBCT Manual Annotation			
Dice similarity coefficient (DSC, %)	mean	94.15 ± 0.68	90.16 ± 2.23	3.99	10.402	0.000 **
	background	99.96 ± 0.01	99.93 ± 0.02	0.03	10.387	0.000 **
	Left MM	91.56 ± 0.97	84.75 ± 4.10	6.81	9.366	0.000 **
	rightMM	90.94 ± 1.33	85.82 ± 3.25	5.12	9.302	0.000 **
Average Hausdorff distance (aHD, mm)	mean	3.68 ± 1.01	5.41 ± 1.63	−1.73	−5.274	0.000 **
	leftMM	3.22 ± 1.01	5.28 ± 2.34	−2.06	−4.791	0.000 **
	rightMM	4.14 ± 1.68	5.54 ± 1.56	−1.40	−3.480	0.002 **

SD, standard deviation. ** *p* < 0.01.

**Table 4 jcm-12-00055-t004:** The average time (seconds, s) cost of segmentation on one case.

Mean ± SD (95%CI)			
Manual Segmenting CBCT	Manual Segmenting CT	Model Segmenting CBCT	Model Segmenting CT
1879.80 ± 338.80 (1753.49, 2006.51)	2245.80 ± 531.72 (2047.46, 2444.54)	5.64 ± 0.63 (5.29, 5.99)	6.76 ± 0.76 (6.34, 7.18)

**Table 5 jcm-12-00055-t005:** The DSC (%) and aHD (mm) of five CBCT and five CT images and the fraction of manual correction (%) between the original auto-segmentations and expert-revised counterparts.

	**Mean ± SD (95%CI)**									
Moda-lity	Dice Similarity Coefficient (DSC, %)				Average Hausdorff Distance (aHD, mm)			Revision (%)		
	Mean	Back-ground	Left MM	Right MM	Mean	Left MM	Right MM	Mean	Left MM	Right MM
CBCT	99.84 ± 0.06 (99.76, 99.92)	100 (100,100)	99.77 ± 0.10 (99.64, 99.90)	99.77 ± 0.19 (99.53, 100)	0.95 ± 0.34 (0.53, 1.37)	0.90 ± 0.36 (0.46, 1.35)	1.00 ± 0.53 (0.34, 1.66)	0.56 ± 0.30 (0.20, 0.93)	0.68 ± 0.66 (−0.14, 1.51)	0.44 ± 0.41 (−0.07, 0.96)
CT	99.84 ± 0.20 (99.59, 100.09)	100 (100,100)	99.72 ± 0.28 (99.38, 100.07)	99.81 ± 0.39 (99.33, 100.29)	0.88 ± 1.27 (−0.70, 2.46)	1.02 ± 0.93 (−0.13, 2.17)	0.75 ± 1.68 (−1.33, 2.83)	0.49 ± 0.59 (−0.24, 1.21)	0.72 ± 1.07 (−0.60, 2.05)	0.25 ± 0.50 (−0.37, 0.86)
Mean ± SD	99.84 ± 0.14 (99.72, 9.94)	100 (100,100)	99.75 ± 0.20 (99.60, 99.89)	99.79 ± 0.27 (99.58, 99.99)	0.92 ± 0.88 (0.29, 1.55)	0.96 ± 0.66 (0.48, 1.44)	0.87 ± 1.18 (0.03, 1.72)	0.52 ± 0.44 (0.21, 0.84)	0.70 ± 0.84 (0.10, 1.30)	0.35 ± 0.44 (0.03, 0.66)

SD, standard deviation; MM, masseter muscle.

## Data Availability

The data presented in this study are available on request from the corresponding author. The data are not publicly available due to privacy limitations.

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
