# Peer review of "Automatic Masseter Muscle Accurate Segmentation from CBCT Using Deep Learning-Based Model"

_jcm, 2022, doi:10.3390/jcm12010055_

Round 1
Reviewer 1 Report
1- Introduction section is informative and covers the literature and clearly stating the objectives of the study.
2- Methods
- CBCT Scans can be taken with large field of view to cover the cranial base. It is unclear to me why the images were taken with a small field of view. Also, why not using CBCTS with large field of views before the surgery.
- It is unclear if authors used the exported t2 with orientation or the CBCT segmentations
Discussion and conclusions:
I suggest that the authors should highlight the clinical significance of MM segmentations to clinicians.
Reviewer 2 Report
This manuscript is excellent and is an important addition to the literature. I enjoyed reading and critically evaluating this manuscript. However, a few points are needed to consider in this article,
Abstract: This part is perfect regarding the study's topic, result, and conclusion. Keywords: simple for readers to know the content of the study.
Introduction: The introduction provides a good, generalized background of the topic that quickly gives the reader an appreciation of the validated deep learning-based automatic segmentation model for CBCT under the refinement of paired CT. However, to make the introduction more substantiated, the authors may add more references related to the need of the study, and, more specifically, about the models, designed before in relation to CBCT’s deep learning-based models.
Material and method: This study has considered every important point required. Methods such as CBCT superimposition on CT, manual annotation of Masseter Muscle, and MM auto-segmentation, implementation details are clear and well explained.
Results: well-written results and tables are self-explanatory. Data are well presented, and no need for any supplementary figures or tables.
Discussion: More in-depth discussion of the result is required.
The conclusion is short but relevant to the study.
References are appropriately marked, and no duplication is seen.
